# Structural insights into IL-11-mediated signalling and human *IL6ST* variant-associated immunodeficiency

Scott Gardner [1,4], Yibo Jin [1,4], Paul K. Fyfe [2,4], Tomas B. Voisin [1], Junel Sotolongo Bellón [3], Elizabeth Pohler[2], Jacob Piehler [3], Ignacio Moraga [2]✉ & Doryen Bubeck [1]✉

IL-11 and IL-6 activate signalling via assembly of the cell surface receptor gp130; however, it is unclear how signals are transmitted across the membrane to instruct cellular responses. Here we solve the cryoEM structure of the IL-11 receptor recognition complex to discover how differences in gp130-binding interfaces may drive signalling outcomes. We explore how mutations in the *IL6ST* gene encoding for gp130, which cause severe immune deficiencies in humans, impair signalling without blocking cytokine binding. We use cryoEM to solve structures of both IL-11 and IL-6 complexes with a mutant form of gp130 associated with human disease. Together with molecular dynamics simulations, we show that the disease-associated variant led to an increase in flexibility including motion within the cytokine-binding core and increased distance between extracellular domains. However, these distances are minimized as the transmembrane helix exits the membrane, suggesting a stringency in geometry for signalling and dimmer switch mode of action.

Cytokines coordinate surface receptors to activate signalling[1–5]. We have a good biophysical and structural understanding of how cytokines engage their receptors[6–19] that explain how changes in cytokine-receptor binding parameters are sensed by cells to initiate specific signalling programs[1,20–22]. These observations suggest that cytokine receptors can behave like dimmer switches and adjust their signalling profiles in response to different environmental cues. However, how cues from cytokine binding are transmitted across receptor extracellular and transmembrane domains to initiate Janus Kinase (JAK) activation and specific signalling responses is unknown[1]. Similarly, the role of cytokine-receptor binding geometry in functional diversification remains the subject of intense debate in the field[1,23,24].

The signalling receptor gp130 represents a paradigm for differential signal activation and functional diversity. It is required for interleukin (IL)−6, IL-11, IL-27, oncostatin-M (OSM), leukaemia inhibitory factor (LIF), ciliary neurotrophic factor (CNF), cardiotrophin-1 (CT-1), and cardiotrophin-like cytokine factor-1 (CLCF1) signalling[25–29]. These cytokines engage specific ligand-binding subunits (often non-signalling), but all require gp130 to control signal transduction. For IL-6 and IL-11 cytokines, gp130 signals by homodimerization of its intracellular domains. Similar to IL-6, IL-11 coordinates gp130 homodimers by forming a hexameric signalling complex comprised of two molecules each of gp130, IL-11 and co-receptor, IL-11Rα. By contrast, signalling in response to stimulation by either IL-27, OSM, LIF, CNF, CT-1 or CLCF1 requires a heterodimeric complex between gp130 and a co-receptor coordinated by a single copy of the cytokine (Supplementary Fig. 1). Activation of gp130 promotes immune regulation, tissue homeostasis, regeneration, and metabolism[30]. While these processes typically involve cytokine signalling in cell types of different origins, individual members of the gp130 cytokine family often elicit contrasting biological activities when acting on the same cell population[25,26,31–33]. How binding of different cytokines is sensed by

[1]Department of Life Sciences, Sir Ernst Chain Building, Imperial College London, London SW7 2AZ, UK. [2]Division of Cell Signalling and Immunology, School of Life Sciences, University of Dundee, Dundee, UK. [3]Department of Biology/Chemistry and Centre for Cellular Nanoanalytics, Osnabrück University, Osnabrück, Germany. [4]These authors contributed equally: Scott Gardner, Yibo Jin, Paul K. Fyfe. ✉e-mail: imoragagonzalez@dundee.ac.uk; d.bubeck@imperial.ac.uk

gp130 and translated into differential signalling and functions remains poorly defined. Work from our laboratory has shown that manipulation of IL-6:gp130 binding kinetics results in activation of biased signalling responses by IL-6 and in decoupling of IL-6 pleiotropic activities[20], highlighting the critical role that the cytokine-gp130 complex stability plays in defining gp130 activities.

Recent studies have reported homozygous mutations in the *IL6ST* gene (encoding for gp130) that result in severe immune deficiencies and bone defects in patients[34–36]. These disease-associated point mutations, N404Y, P498L and A517P, result in loss of function for some but not all gp130 cytokines. Of these, N404Y abrogates signalling by IL-6, IL-11, IL-27 and OSM, while the LIF response remains unchanged[35]. The mutation P498L, however, causes a reduction in response to stimulation by IL-6 and IL-27 in peripheral blood mononuclear cells and IL-6 and IL-11 stimulation in fibroblasts, indicating signalling effects can also be influenced by cell type[35]. All three of these disease-associated mutations dramatically disrupt signalling when engaged by cytokines that homodimerize gp130 signalling domains, such as IL-6 and IL-11, but still retain some degree of signalling when bound by cytokines that bind gp130 in heterodimeric complexes, e.g. LIF and IL-27 (Supplementary Fig. 1)[34]. These mutations do not affect gp130 expression; P498L and N404Y gp130 variants are trafficked to the surface at similar levels to wildtype[37]. All of these disease-affected residues map to hinge regions between gp130 fibronectin domains, away from the cytokine binding domains (D1-D3) (Supplementary Fig. 1). Interestingly, gp130 domains D4 and D5 associate in an "elbow-like" fold[38], that it is critical for gp130-mediated signal transduction[39–41]. Thus, it is possible that mutations in these domains could affect the stability, geometry or flexibility of cytokine-gp130 complexes and impact downstream signalling. However, the molecular and structural basis for how these mutations disrupt gp130 activities and generate cytokine specificity remains unknown. Since these gp130 mutations are predicted to not affect cytokine binding, but still result in loss of function, they represent a unique opportunity to understand how cytokine-receptor binding is transmitted across the plasma membrane into specific signalling programs and bioactivities.

Here we set out to investigate the molecular basis underpinning the diversity in gp130 signalling outcomes in health and disease. Using cryoEM we define interaction interfaces of IL-11, its co-receptor IL-11Rα and gp130 that differ from other cytokines and reveal structural rearrangements in IL-11 that occur upon receptor binding. We explore how disease affected residues in gp130 impact cytokine-receptor complex stability, stoichiometry and topology. Our data show that the human disease affected gp130 variant P498L (*murine* equivalent gp130$_{P496L}$) contributes to flexibility of the receptor ectodomain and influences motions across both the IL-11 and IL-6 hexameric core complexes. Through molecular dynamics simulations of the IL-6 transmembrane receptor complex, we discover that although the gp130 disease variant causes large differences in the distance between extracellular domains, these differences are diminished for membrane-proximal intracellular residues.

## Results and discussion
Several cytokines use gp130 to relay signals across the plasma membrane. To understand how gp130 acts in health and disease states, we generated IL-6 and IL-11 receptor recognition complexes with gp130 variants defective in pSTAT3[37]. To overcome challenges in complex stability, we engineered human IL-6 and IL-11 cytokines fused by a short linker to the second and third domains of their co-receptors, IL-6Rα and IL-11Rα respectively (Supplementary Fig. 2a). The fusion cytokine was expressed in insect cells and purified together with domains D1-D6 of gp130. Due to low yields when purifying the *human* gp130 variant, *murine* wildtype gp130 or gp130$_{P496L}$ (*human* equivalent P498L) (Supplementary Fig. 2) was used for structural studies and binding assays.

IL-11 signals by binding both its co-receptor, IL-11Rα, and gp130. While the cytoplasmic domains of gp130 recruit JAKs and effector proteins responsible for downstream signalling, IL-11Rα is necessary for ligand binding to coordinate the hexameric assembly that defines the orientation of gp130 signalling domains. To understand how IL-11 instructs unique signalling outcomes through gp130, we solved the structure of the IL-11 receptor recognition complex by cryoEM (Fig. 1 and Supplementary Fig. 2c and Supplementary Fig. 3). Complexes comprised two copies of the human IL-11 cytokine variant (IL-11 fused to domains D2 and D3 of IL-11Rα) and two copies of *murine* gp130$_{P496L}$ mutant (D1-D6), in agreement with a *human* IL-11 complex which came into press after submission of our manuscript[42]. We used the ab initio reconstruction protocols within cryoSPARC[43] to generate an initial model of the complex. Maps were further refined using a combination of nonuniform and heterogenous refinement procedures. The final map was refined to a reported resolution of 3.1 Å using the gold standard FSC 0.143 cut-off; with local resolution ranging from 2.6 Å to 5.4 Å. Initial models for the complex were built using AlphaFold2 predictions for IL-11 and gp130 (D1-D6). Chirality of the IL-11 helical bundle was used to assign the handedness of the reconstruction. Models were refined with iterative building and refinement of side chains where density permitted (Supplementary Table 1).

IL-11 is a helical cytokine that coordinates a hexameric assembly with two copies of IL-11, IL-11Rα and gp130 through interactions at three principal binding sites (Fig. 1a, b). Site 1 of IL-11 is formed by the hinge between domains D2 and D3 of IL-11Rα (Fig. 1c). Aromatic residues of IL-11Rα (F187, Y125, F252, W188, F298) together with H251 and L299 define a geometry at the apex of the elbow between these two domains specifically recognized by IL-11:R190. In our complex IL-11 residues T77-L90 are at an interface with IL-11Rα and adopt a helical conformation (Fig. 2a). A comparison with the crystal structure of apo IL-11 shows that these residues undergo a conformational change upon complex formation (Fig. 2b, d). IL-11 binding to its co-receptor IL-11Rα is further stabilized by additional interaction interfaces with two copies of gp130 within the hexameric assembly, referred to as Site 2b and Site 3b (Fig. 1b).

Similar to the hexameric IL-6 complex[44], both IL-11 and IL-6 engage one copy of gp130 at Site 2a. To understand the molecular basis for differences in IL-6 and IL-11 signalling, we investigated structural differences at gp130 interaction interfaces. To ensure that any variations in topology were not due to the human/mouse chimeric complex used in this study, we additionally solved the cryoEM structure of the equivalent chimeric variant of the IL-6 receptor recognition complex (Supplementary Fig. 2, Supplementary Fig. 4 and Supplementary Table 1). Superposition of our human/mouse chimeric IL-6 complex with the *human* IL-6 hexameric assembly[44] based on gp130 had an RMSD of 1.1 Å. Interface residues are conserved between the two species (Supplementary Fig. 2), thus we conclude that any potential differences due to the chimera were negligible. We observe that for the IL-11 complex, the sidechain of IL-11:R135 at Site 2a inserts into a hydrophobic groove created by a hinge between domains D2 and D3 of gp130 (V190, W164, F169, Y184, I192, *murine* numbering) (Fig. 1d). The sidechains of several arginine residues along the same helix of IL-11 (R132, R135, R138, R139) further coordinate the orientation of gp130 at Site 2a (Fig. 1d). Critically, none of these arginine residues are conserved in IL-6 at the Site 2a interface (Fig. 3a).

The orientation of gp130 at Site 2a is stabilized by electrostatic interactions between residues of IL-11Rα with domain 3 (D3) of gp130, referred to as Site 2b (Fig. 1e). Here, IL-11Rα:R235 forms a salt bridge with gp130:E273. By contrast, gp130:E273 points towards a tryptophan (W233) in the IL-6 receptor recognition complex (Fig. 3b). Across the extended Site 2b interface, additional salt bridge interactions are formed between gp130:R279 and an aspartate residue in both the IL-6 and IL-11 receptor recognition complexes (IL-6Rα:D281 and IL-11Rα:D282, respectively) (Fig. 3b); however, mutation of the *human*

equivalent gp130 arginine (R281Q) selectively impairs signalling by IL-11 but not by IL-6[1,44,45]. Our structural data show that local environment for this conserved interaction varies between the two co-receptors. For IL-6Rα, the switch from arginine to glutamine could be accommodated by IL-6Rα:H280. In contrast, the D3 gp130-IL-11Rα interaction interface is dominated by polar residues (IL-11Rα:Y260, S269, T281), which could explain the selective loss of function for IL-11 signalling. Our data are consistent with molecular dynamics simulations of IL-11 homology models that showed impaired interactions of this mutation with IL-11Rα[45] and suggest a crucial role of arginine residues within the IL-11 receptor recognition complex that may encode subtle variations in gp130 orientation driving differences in IL-6 and IL-11 signalling outcomes.

Within the hexameric assembly, a second copy of gp130 binds to Site 3 of IL-11, referred to as Site 3a (Fig. 1f). Analogous to the Site 3 interactions observed for both IL-6[44,46] and IL-27[17-19,46], gp130 engages IL-11 through the leading edge of its D1 β-strand with subtle differences in interface residues. In the IL-11 complex, the Site 3a is comprised of a series of electrostatic interactions (gp130:K68-IL-11:D67, gp130:H71-IL-11:D62) and pi stacking between IL-11:W168 and gp130:Y116 (Figs. 1f, 3c). The equivalent aromatic residue in the helical cytokine component of IL-27 (p28:W195) also forms a pi stacking interaction with gp130:Y116 in the IL-27 receptor recognition complex[17-19,46] and plays a key role in signalling[47]. Although the N-terminus of gp130 is largely disordered in our IL-11 complex, these residues form an additional β-strand that sandwiches IL-6 and may contribute to differences in gp130 orientations between the two cytokine complexes (Fig. 3c). Binding of gp130 at Site 3a is additionally stabilized by a second interface, Site 3b, which coordinates sidechains across all three components of the complex (IL-11, IL-11Rα and gp130) (Figs. 1g, 3d).

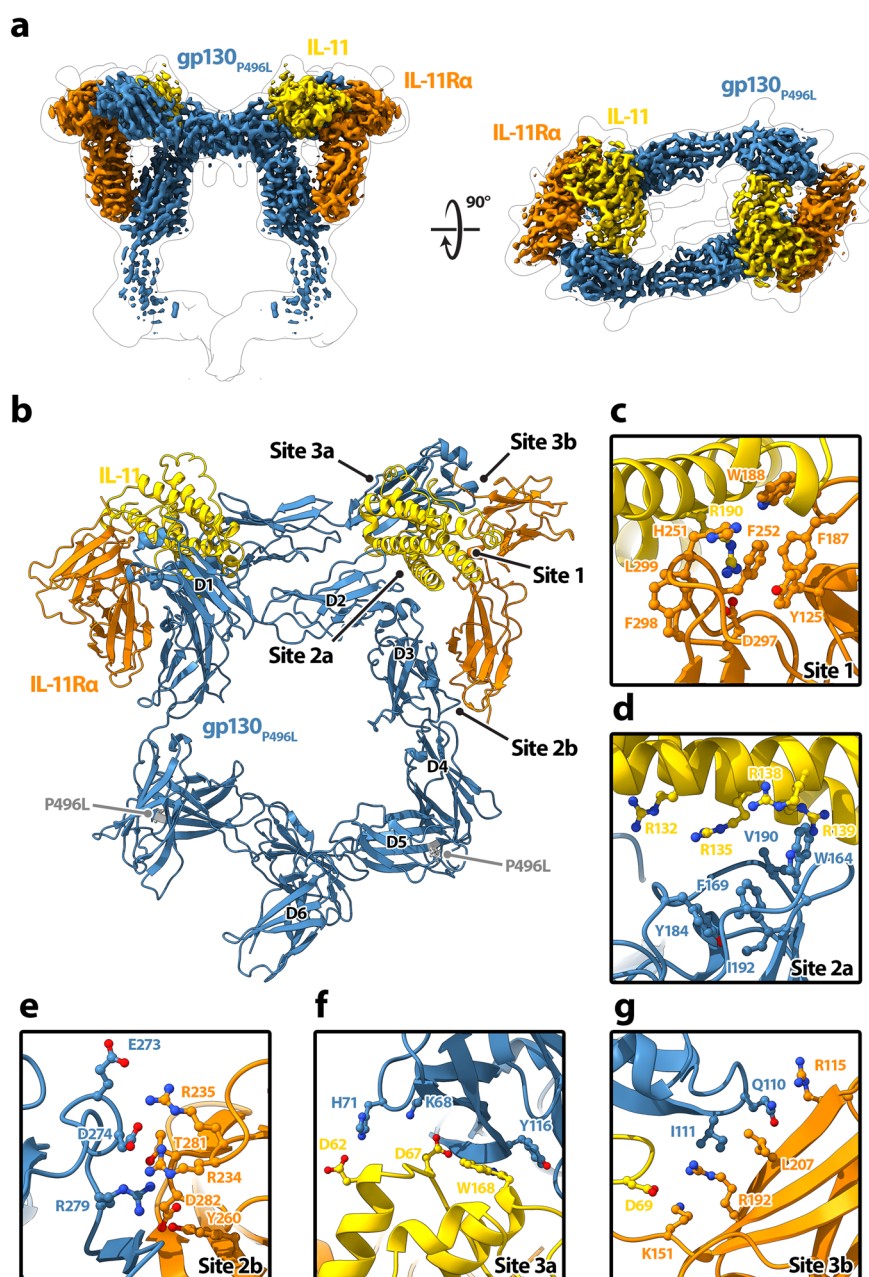

**Fig. 1 | Structure of the IL-11 receptor recognition complex.** CryoEM structure (**a**) and atomic model (**b**) of the IL-11 receptor recognition complex. The unsharpened map is shown as transparent silhouette in (**a**). Ribbon representations of gp130P496L (blue), IL-11 (yellow), IL-11Rα (orange) are shown. Interaction interfaces for the complex at Site 1 (**c**), Site 2a (**d**), Site 2b (**e**), Site 3a (**f**) and Site 3b (**g**) are shown. Sidechains for interface residues are represented as sticks. Domains 1-6 and Cα position of residue 496 (grey) of gp130P496L are indicated in panel (**b**).

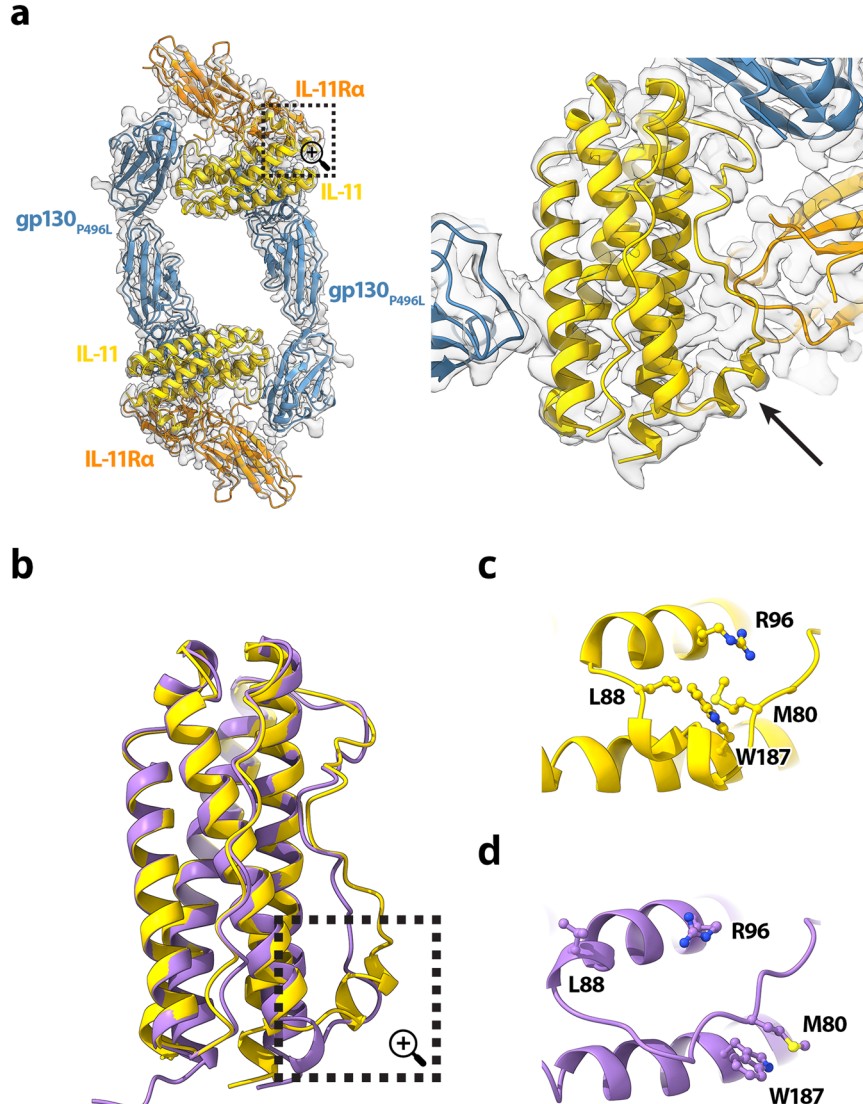

**Fig. 2 | Comparison of apo and bound IL-11.** Structural model of the IL-11 receptor recognition complex (ribbon representation) overlaid on the cryoEM map (transparent surface) (**a**). Ribbon representations of gp130 $_{P496L}$ (blue), IL-11 (yellow), IL-11Rα (orange) are shown. The region encompassed by the dotted box is enlarged in the right panel. Arrow indicates residues T77-L90 of IL-11. **b** Superposition of IL-11 from the IL-11 receptor recognition complex (yellow) with the apo IL-11 structure (purple, PDB:6O4O); Residues T77-L90 are boxed. This region is shown in detail for the bound form of IL-11 (**c**) and apo IL-11 (**d**).

Mutations in the gene *IL6ST* encode gp130 variants that impact both IL-6 and IL-11 signalling. The affected residues (A517P, N404Y and P498L in gp130) occur at hinge regions between domains of gp130 (Supplementary Fig. 1). Molecular dynamics simulations of gp130 ectodomains for each of these variants showed increased mobility between pairs of domains adjacent to the mutation analysed. In these experiments, changes caused by N404Y and A517P were more pronounced than those observed for P498L[37]. As these simulations were performed with individual gp130 ectodomains, it remained unclear what the impact of mutation P498L would be in the context of trans-membrane gp130-cytokine complexes.

Although gp130$_{P498L}$ mutation is predicted to not affect IL-6 and IL-11 binding, formal evidence remains lacking. Thus, we set to investigate whether gp130$_{P498L}$ mutation affects IL-6 and IL-11 binding or receptor assembly, which could account for its poor signalling. Due to poor purification yields of the *human* gp130 variant, we defined the binding affinities of IL-6 and IL-11 for recombinant *murine* wildtype gp130 and gp130$_{P496L}$ via surface plasmon resonance (SPR). We immobilized biotinylated wildtype gp130 or gp130$_{P496L}$ on a strepta-vidin SPR surface and passed a range of IL-6 or IL-11 concentrations to

measure rates of association and dissociation (Supplementary Fig. 5a). We found that both IL-6 and IL-11 bound with similar affinities to wildtype gp130 or gp130$_{P496L}$ (Supplementary Fig. 5). We next wanted to investigate the role the disease-associated residue in cell signalling. To ensure native interactions with downstream signalling partners, we transfected a HeLa cell line lacking endogenous gp130 with DNA encoding *human* wildtype gp130 or gp130$_{P498L}$. HeLa cells expressing gp130$_{P498L}$ induced significantly lower levels of STAT3 phosphoryla-tion in response to IL-6 and IL-11 stimulation than cells expressing wildtype gp130 (Supplementary Fig. 6a), confirming that poor signal-ling by these gp130 mutants is not the result of weak cytokine binding.

We have shown that differences in kinetics of gp130 hexameric complex assembly can result in altered signalling[20]. To explore whe-ther similar principles underly altered signalling by gp130$_{P498L}$, we probed the assembly of gp130 dimers at the single molecule level using dual-colour total internal reflection fluorescence (TIRF) microscopy using the same engineered HeLa cell line as the signalling assays. In these experiments, wildtype gp130 or gp130$_{P498L}$ were N-terminally tagged with a meGFP, which was rendered non-fluorescent by the Y67F mutation (Supplementary Fig. 6b). This tag (mXFP) is recognized by

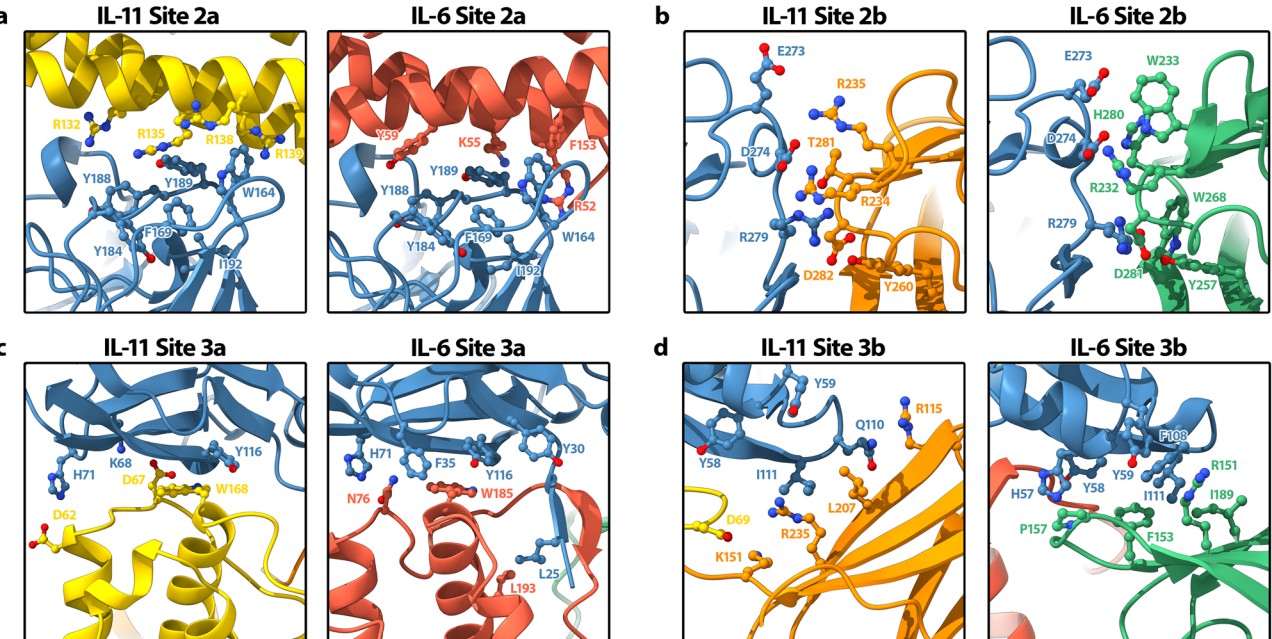

**Fig. 3 | Structural comparison of the IL-11 receptor recognition complex with the IL-6 cytokine complex.** Interaction interfaces of IL-11 (left) or IL-6 (right) with gp130$_{P496L}$ (blue) at Site 2a (**a**), Site 2b (**b**), Site 3a (**c**) and Site 3b (**d**). IL-11 (yellow), IL-11Rα (orange), gp130$_{P496L}$ (blue), IL-6 (coral), IL-6Rα (green) are shown as ribbons with sidechains of interface residues shown as sticks. The IL-6 structural model shown is derived from the chimeric *human* cytokine IL-6 in complex with *murine* gp130$_{P496L}$ for a direct comparison with the IL-11 complex solved in this study.

dye-conjugated anti-GFP nanobodies (NB), allowing selective fluorescence labelling of gp130 at the cell surface of live cells. Single-molecule co-localization and co-tracking analysis was used to identify correlated motion (co-diffusion) of the two spectrally separable fluorophores, which was taken as readout for gp130 dimerization. In absence of ligand stimulation, no significant gp130 dimer levels were observed (Supplementary Fig. 6c). After IL-6 and IL-11 stimulation, strong gp130 dimerization was found, with very similar levels observed for wildtype gp130 and gp130$_{P498L}$. Likewise, stimulation with IL-6 and IL-11 induced similar changes in the diffusion constants of wildtype gp130 and gp130$_{P498L}$ (Supplementary Fig. 6d). These observations confirm that the P498L mutation neither affects cytokine binding, dimerization efficacy nor dynamics, yet impairs downstream signalling. However, there may be effects on dynamics or binding kinetics that could additionally impact signalling when the cytokine is not fixed in a single chain with its co-receptor.

To investigate the structural basis for impaired signalling due to gp130$_{P498L}$, we characterized conformational flexibility of our soluble cytokine recognition complexes. We used a deep neural network to model continuous flexibility[48] of the IL-11 receptor recognition complex with the equivalent *murine* variant, gp130$_{P496L}$ (Supplementary Movie 1). We observe motions within the disease affected hinge region that result in less ordered density for gp130 domains D5 and D6. In addition, our analysis shows flexibility across the hexameric core. Although a reported cryoEM structure of *human* IL-11 in complex with wildtype gp130 showed flexibility for the C-terminal domains of gp130[42], we were unable to directly assess the impact of the disease variant due to unavailability of structural models when we submitted our manuscript. To test our hypothesis that the disease variant introduces more flexibility within the complex, we directly compared the extent and directions of motions within IL-6 complexes comprised of either wildtype gp130 (Supplementary Fig. 7) or gp130$_{P496L}$ variant (Supplementary Fig. 4 and Supplementary Movie 2). For cytokine complexes with gp130$_{P496L}$, we report two dimensions of latent motion. Scatter plots showing the final distribution of particle latent coordinates

are consistent with continuous motion, rather than discrete conformational states (Supplementary Movie 2). There are two dominant moving parts corresponding to the two gp130 molecules, which extend from a more rigid central hexameric core (Supplementary Movie 2). Superposition of a series of convected densities from the 3Dflex model along the first latent dimension show a high degree of mobility about the affected gp130:P496L residue for both the IL-6 and IL-11 complexes (Supplementary Movies 1 and 2). Though more pronounced for the IL-6:IL-6Rα:gp130 variant complex (Supplementary Movie 2), this flexibility is propagated throughout both structures with additional flexing occurring between the two trimers of the hexameric core. Together our structural analysis shows that for mutant complexes, the magnitudes of motions are larger and domains D5 and D6 of gp130 are less well resolved compared to wild type.

Although our structural analyses of ectodomain complexes were consistent with molecular dynamics simulations (MD) of soluble gp130 variants[37], it remained unclear what role the membrane has on the flexibility of the complex or on the orientation of gp130 transmembrane helices. To address this, we next ran atomistic MD simulations for IL-6/IL-6Rα/gp130 models within a lipid membrane. Based on Alphafold2 structural predictions, we created models for *human* wildtype gp130 and gp130$_{P498L}$ truncated after the transmembrane and juxtamembrane helixes. Given that the intracellular regions of IL-6Rα are not required for signalling and were not present in our cryoEM studies, only the cytokine-binding domains of IL-6Rα are included in our simulations. We next analysed how the orientation of the two gp130 transmembrane helices varied throughout the simulation. We measured distances between pairs of residues located at the extracellular (E617), central (L630), and intracellular (N642) regions of the transmembrane helix (Fig. 4). Our simulations of the wildtype complex show that the extracellular face of these helixes is separated by an average distance of 24.8 Å (Fig. 4), consistent with the cryoEM structure of the transmembrane IL-6:IL-6Rα:gp130 complex[46]. However, the average separation distance for the variant is much larger (35.6 Å) with more variation about the mean distance

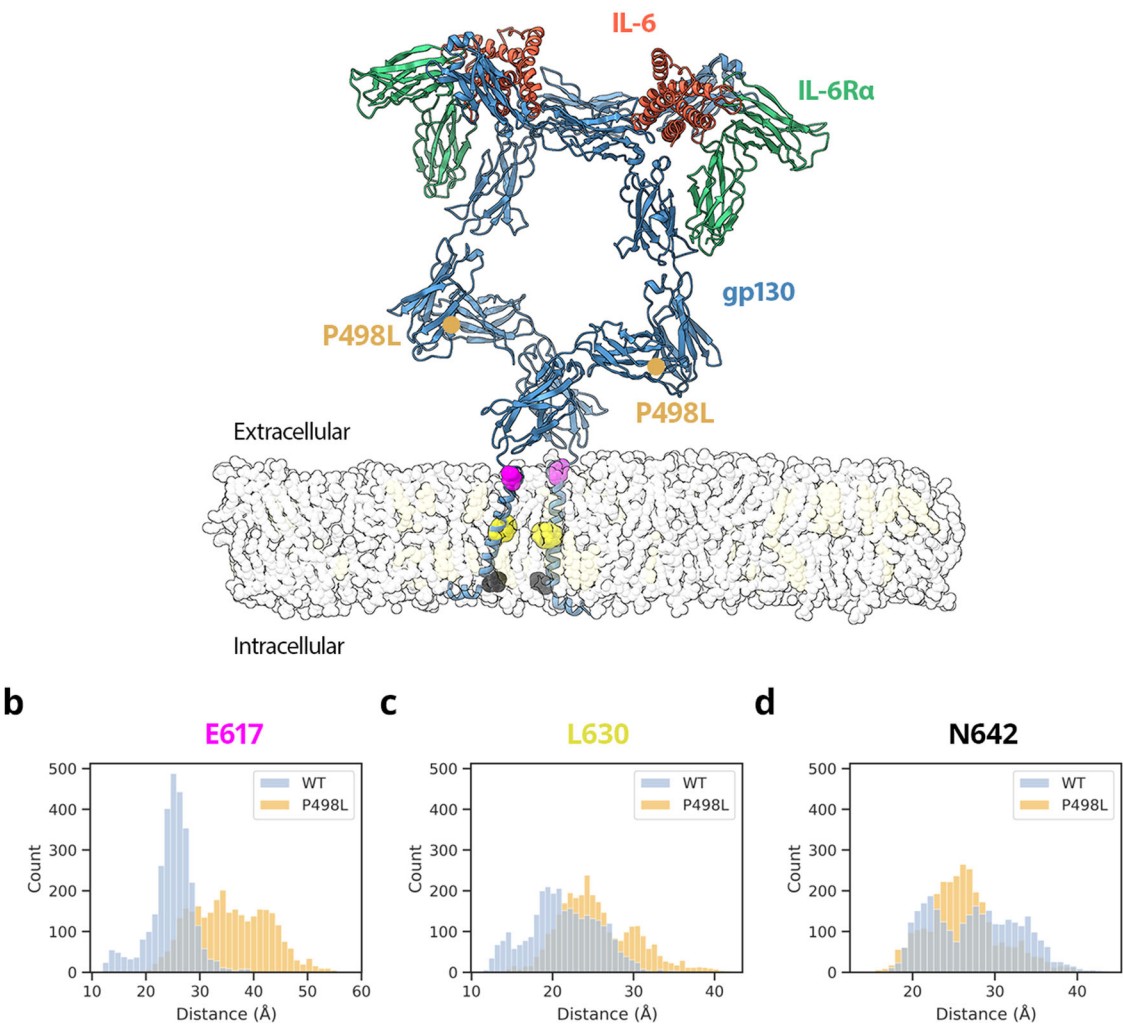

**Fig. 4 | Dynamics and molecular interactions of human disease associated mutation gp130$_{P498L}$ on the transmembrane IL-6 complex.** Atomistic model of the wildtype *human* IL-6 complex derived from AlphaFold2 models of IL-6 (coral) residues:47-212 IL-6Rα residues:115-315 (green) and gp130 residues:28-653 (blue) embedded in a lipid bilayer (transparent) containing POPC (grey) and cholesterol (light yellow). Location of gp130 disease-affected residue P498L is shown as a tan circle. Within the transmembrane helix of gp130, E617 (pink), L630 (yellow) and N642 (black) are shown as spheres (**a**). Histograms show distribution of pairwise Cα carbon distances in Å between gp130 residues E617 (**b**), L630 (**c**) and N642 (**d**), for three independent simulations of wildtype (blue) and P498L variant (orange) complexes. Counts refer to the number of frames of the simulation where the distance between pairs of atoms corresponds to the value indicated on the X axis.

(7.0 Å). Unexpectedly, this difference in separation distance is not maintained throughout the length of the helix. At the mid-point of the transmembrane helix (L630), the average distance separating the wildtype helices is 21.3 Å, while the mutant is separated by 25.5 Å (Fig. 4). As the helix approaches the intracellular face (N642), the difference in average distances is less than 2 Å (27.6 Å and 26.2 Å for wildtype and mutant complexes, respectively). These data suggest there may be subtle differences between the mutant and wildtype complexes that contribute to impaired signalling outcomes. As our simulations only included the minimal domains of IL-6Rα required for signalling, it may be possible that the full-length co-receptor could additionally influence the orientation of gp130 transmembrane helices.

Previous studies report that the impact on signalling for the P498L variant was less pronounced than for N404Y and A517P disease affected residues[37]. Our data further support a model for whereby small differences in intracellular distances can serve as dimmer switches in gp130 signalling. An important characteristic shared by receptors of the gp130 family, i.e. gp130, LIFR, and IL-27Rα, is the presence of an acute bend between their third-to-last and second-to-last extracellular domains (namely, gp130 D4D5, LIFR D6D7, and IL-27Rα D3D4), which is believed to play a role in signalling and where gp130 mutations are localized. A recent study suggests that this bend serves to bring the bottom centres of the receptor juxtamembrane domains to within about 30 Å, a critical step to initiate signalling[46]. Together, our data suggest how the disease associated gp130 variant P498L can still illicit some, albeit impaired signalling.

The cell surface receptor gp130 decodes several cytokine cues to relay signalling messages across the plasma membrane. Our cryoEM structure of the gp130:IL-11 receptor recognition complex, together with structural, cellular and computational experiments of cytokine complexes with gp130 disease variants provide insight into immune regulation. Our data explain how subtle variations in interaction interfaces drive differences in signalling outcomes by altering the conformational dynamics of the gp130 signalling complex. In conclusion, our study provides a molecular framework to understand how mutations in surface receptors that do not affect ligand binding influence signalling and may inform new strategies for correcting defective signalling by these mutated receptors.

## Methods

### Protein expression and purification

HypIL-6 and HypIL-11 were cloned as a linker-connected single-chain variant (IL-6 + IL-6Rα; IL-11 + IL-11Rα) as described in[49,50] (Supplementary Fig. 2). In the text, we refer to this variant only as IL-6 and IL-11. DNA sequences encoding *murine* wildtype gp130 and gp130$_{P496L}$ were cloned into the pAcGP67-A vector (BD Biosciences) in frame with an N-terminal gp67 signal sequence, driving protein secretion, and a C-terminal hexahistidine tag. Baculovirus stocks were produced by transfection and amplification in *Spodoptera frugiperda* (Sf9) cells grown in SF900III media (Invitrogen), and protein expression was carried out in suspension *Trichoplusia ni* (High Five) cells grown in InsectXpress media (Lonza). Hi-Five cells were pelleted with centrifugation at $1000 \times g$, and impurities from the remaining media were removed by a precipitation step through addition of Tris pH 8.0, $CaCl_2$, and $NiCl_2$ to final concentrations of 200, 50, and 1 mM, respectively. The precipitate formed was then removed through centrifugation at $9000 \times g$. Nickel-NTA agarose beads (Qiagen) were added to the clarified media and the target proteins purified through batch binding followed by column washing in HBS buffer. Elution was performed using HBS buffer plus 200 mM imidazole. Final purification was performed by size exclusion chromatography on an ENrich SEC 650 10 × 300 column (Biorad), equilibrated in HBS or HBS-Hi buffers. Concentration of the purified sample was carried out using 30 kDa Millipore Amicon-Ultra spin concentrators. Recombinant proteins were purified to greater than 98% homogeneity. For cryoEM studies, gp130 (wildtype or P496L), and IL-6/IL-11 were mixed in a 1:1 ratio and subsequently purified by size-exclusion chromatography.

To generate biotinylated proteins for surface plasmon resonance studies, the gp130 sequence was subcloned into the pAcGP67-A vector with a C-terminal biotin acceptor peptide (BAP)–LNDIFEAQKIEWHW followed by a hexa-histidine tag. Proteins with a C-terminal biotin acceptor peptide were biotinylated in vitro using the soluble BirA ligase enzyme in 0.5 mM Bicine, pH 8.3, 100 mM ATP, 100 mM magnesium acetate, and 500 mM biotin (Sigma)[20].

### Surface plasmon resonance

Surface plasmon resonance experiments were performed to determine the binding affinity of the recombinantly produced IL-6 and IL-11 to *murine* wildtype gp130 and gp130$_{P496L}$. These were carried out on a Biacore T100 instrument (T200 sensitivity enhanced). C-terminal biotinylated wildtype gp130 and gp130$_{P496L}$ were immobilized onto a SA sensor chip (GE Healthcare) at levels of ~100 response units (RU). The immobilization was performed in 10 mM HEPES, 150 mM NaCl, 0.02% (v/v) TWEEN-20, pH 7.2 buffer. Analysis runs were performed at 25 °C in 10 mM HEPES, 150 mM NaCl, 0.05% (v/v) TWEEN-20, pH 7.2, and 0.5% BSA. Data analysis was performed using Biacore T200 Evaluation Software v3.0.

### Phospho-flow analysis

For phospho-flow analysis of STAT3, gp130 KO HeLa cells[20] were transfected with *human* gp130 wildtype or P498L mutant and simulated with a saturated concentration (100 nM) of IL-6 or IL-11 for 15 min at 37 °C before fixation with 2% paraformaldehyde for 10 min at room temperature. HeLa gp130 KO cells were derived from HeLa cells obtained from the German Collection of Microorganism and Cell Cultures GmbH (ACC 57). Cells were washed in PBS and permeabilized in ice-cold 100% methanol and incubated on ice for a minimum of 30 min. After permeabilization, cells were fluorescently barcoded using two NHS-dyes (PacificBlue, #10163, DyLight800, #46421, Thermo Scientific). Individual wells were stained with a combination of different concentrations of these dyes[51]. Once barcoded, cells were pooled and stained with anti-pSTAT3Alexa488 (Biolegend #651006) used in a 1/50 dilution. Phospho-flow data was collected using CytoFlex flow cytometer using Kaluza Analysis software v1.3. During acquisition,

individual populations were identified according to the barcoding pattern and pSTAT3Alexa488 MFI was quantified for all populations. MFI was plotted using Prism software v7 (GraphPad).

### Live-cell dual-colour single-molecule imaging studies

Dual-colour single-molecule imaging was carried out by total internal reflection fluorescence microscopy (TIRFM) using an inverted microscope (IX-71, Olympus) equipped with a manual triple-line TIRF condenser (Olympus) and a 150× oil immersion objective (UAPON 150x TIRF, NA 1.45, Olympus) as recently described in detail[52]. The fluorophores Rho11 and AT643 were excited using a 561 nm diode-pumped solid-state laser (max. power 200 mW, Cobolt Jive, Cobolt) and a 643 nm laser diode (max. power 140 mW, LuxX 642-140, Omicron), respectively. Fluorescence was filtered by a quad-band polychroic mirror (zt405/488/561/640rpc, Chroma Technology) and excitation light was blocked by a quad-band bandpass emission filter (BrightLine HC 446/523/600/677, Semrock). Emission light was detected using a single back-illuminated EMCCD camera (Andor iXon3 897, Andor Technology) after passing a two-color image splitter (DualView DV2, Photometrics), which is equipped with a dichroic beamsplitter at 640 nm (640 dcxr, Chroma) and two single-band bandpass emission filters (BrightLine HC 600/37, Semrock, HQ 690/70, AHF).

For single molecule tracking and co-tracking, *human* gp130 WT and gp130$_{P498L}$ fused to an N-terminal mXFPm (mECFP-W66F-E147K-H164N) were employed for selective cell surface labelling via anti-GFP nanobodies 'minimizer'[52], which was generated in house and used at 1.5 nM concentration. For this purpose, the ORFs of gp130 and gp130$_{P498L}$, respectively, lacking the signal peptide were cloned into the vector pSems comprising the signal peptide of Igκ chain followed by mXFP (pSems-leader-mXFPm-gp130 and pSems-leader-mXFPm-Gp130$_{P498L}$). gp130 KO HeLa cells[20] were transfected with pSems-leader-mXFPm-gp130 or pSems-leader-mXFPm-Gp130$_{P498L}$ in 6 cm dishes at 70% confluency using a polyethylenimine (PEI) transfection protocol[52] at the day before imaging. For imaging, transiently transfected cells were seeded on microscopy cover slides coated with a 50/50 (w/w) mixture of poly-L-lysine graft copolymers of polyethylene glycol (PLL-PEG) that were modified with an RGD-peptide and a terminal methoxy group, respectively[53], to block unspecific binding of labelled nanobodies to the cover slide. Microscopy experiments were performed in presence of an oxygen-scavenging system composed of glucose oxidase (4.5 U*mL$^{-1}$), catalase (540 U*mL$^{-1}$), glucose (4.5 mg*mL$^{-1}$), ascorbic acid (1 mM) and methyl viologen (1 mM)[54] to increase photostability. Cell surface mXFPm-gp130 WT and mXFP-gp130$_{P498L}$ were labelled by co-incubation with 5 nM of anti-GFP nanobody minimizer site-specifically conjugated with ATTO Rho11 (Rho11) and ATTO 643 (AT643) maleimide, respectively (ATTO-TEC GmbH). Both nanobodies were kept in solution throughout the experiment to ensure consistent high DOL. Cells were imaged without stimulation and after addition of 50 nM HypIL-6 or HypIL-11. Videos of viable cells were recorded at 30 fps for typically 150 consecutive frames using Andor iQ 2.4.4 software (Andor Technology) for image acquisition. Microscopy image stacks were subjected to single-molecule co-localization and co-tracking analysis using the custom-made MATLAB script SLIMfast 4 C (https://doi.org/10.5281/zenodo.5712332)[52]. Single molecules confined to a radius of 100 nm for more than one second (30 frames) were classified as immobile particles and removed from further calculations. Relative co-diffusion levels were determined for each cell recorded in the experiments and corrected for stochastic dual-colour labelling. Statistical analysis by an unpaired student's *t* test was carried out from typically 15 cells recorded for each condition.

### CryoEM sample preparation and data collection

Lacey carbon Au 300-mesh grids (Electron Microscopy Sciences) were glow discharged in residual air for 60 s using a Cressington 208

(Cressington Scientific Instruments). Four microlitres of 0.1 mg/mL IL-6 (*murine* gp130), IL-6 (*murine* gp130$_{P496L}$), or IL-11 (*murine* gp130$_{P496L}$) was applied to the carbon side of the grid. The grid was blotted and plunge-frozen into liquid ethane using a Vitrobot mark IV (Thermo-Fisher Scientific) operating at 20 °C and 100% humidity, with a blot time of 2 s and a blot force of −2. Grids were stored in liquid nitrogen until use.

Electron micrograph movies were collected using a 300 keV Titan Krios (Thermo Fisher Scientific) fitted with a K3 (Gatan) direct electron detector operating in super-resolution mode. Datasets were collected via EPU v1.12.079 using aberration-free image shift (AFIS) with fringe-free illumination (FFI). Micrograph movies were collected at a nominal magnification of 81,000× and binned by two on the camera. Full data collection details for each data set are supplied in Supplementary Table 1.

### CryoEM data processing

All image processing steps were performed in CryoSPARC v4.2.1[43]. The same overall approach was used to process each data set, and image processing workflows are summarised in Supplementary Fig. 3, Supplementary Fig. 4 and Supplementary Fig. 7. Raw micrograph movies were corrected for beam-induced motion using patch motion correction. CTF parameters of motion-corrected micrographs were estimated using patch CTF. Particles were picked using pretrained models and custom trained models in Topaz v0.2.5[55]. Particles were down sampled by a factor of four and extracted from micrographs. Down-sampled particles were used in 1–2 rounds of 2D classification (number of classes = 100, batch size per class = 1000, number of online-EM iterations = 100, number of final full iterations = 5). Particles from featureless, noisy, or poorly resolved classes were discarded. Particles from well-resolved 2D classes were subjected to ab initio reconstruction (number of classes = 3). Particles from the best resolved ab initio class were subjected to homogenous refinement, then re-extracted from micrographs with two-times down sampling and again used in homogenous refinement. Particles were then re-extracted without down-sampling and used in non-uniform refinement[56] followed by local motion correction[57]. Motion-corrected particles were used for two non-uniform refinements, applying either C1 or C2 symmetry. Per-particle defocus values and per-exposure-group CTF parameters (tilt, trefoil, spherical aberration, tetrafoil, and anisotropic magnification) were refined during non-uniform refinement. Particles refined with C1 symmetry were used for 3D Flexibility analysis[58]. The C2 symmetry reconstructions were filtered by local resolution using the negative B-factor value calculated during non-uniform refinement.

### Model building and refinement

An initial model of the IL-11 complex was built using the AlphaFold predictions of the IL-11 cytokine (AlphaFold Protein Structure Database A8K3F7), IL-11Rα (AlphaFold Protein Structure Database Q14626), and *murine* gp130 (AlphaFold Protein Structure Database Q00560). An initial model of the IL-6 complex was built using the crystal structure of hexameric IL-6 (Protein Data Base 1P9M). We replaced the model of gp130 in the IL-6 structure with the AlphaFold prediction of *murine* gp130 (AlphaFold Protein Structure Database Q00560). Initial models were trimmed to reflect the domain boundaries in the constructs used, then rigid body fitted to locally filtered cryoEM maps using UCSF ChimeraX v1.6.1[59]. The initial model was refined into the density using Isolde v1.6.0[60]. Due to low local map resolution for gp130 D4 – D6, adaptive distance restraints were applied to these domains during refinement in Isolde. Where the density allowed, glycans predicted on Uniprot were built using the carbohydrates module in Coot v0.8.9[61]. Complete models were subjected to real-space refinement in Phenix v1.21[62] using reference model restraints, secondary structure restraints, Ramachandran restraints, and non-crystallographic symmetry (NCS) constraints. Model B-factors were refined in Phenix. Map-

model FSC and full cryoEM validation were assessed using inbuilt validation tools in Phenix. Model validation statistics are provided in Supplementary Table 1. Interface residues were examined using PDBe PISA v1.52 and ClustalW.

### Molecular dynamics simulations

Atomistic models of IL-6 receptor complexes comprised of *human* gp130 and gp130$_{P498L}$ were generated using AlphaFold2. Models were trimmed to remove regions with low pLDDT scores, reflecting low confidence predictions. As transmembrane regions of IL-6Rα are not involved in signalling, this model was further truncated to reflect domains involved in known extracellular interactions. Final boundaries for individual chains include residues:47-212 for IL-6, residues:115-315 for IL-6Rα, and residues:28-653 for gp130. Models were aligned to the crystal structure of the *human* hexameric core complex[44] to generate wildtype IL-6:IL-6Rα:gp130 and IL-6:IL-6Rα:gp130$_{P498L}$ models used in atomistic molecular dynamics (MD) simulations. Simulation boxes were prepared with the Bilayer Builder utility of CHARMM-GUI[63,64] and PPM 2[65] to place each structure in a 22.5 × 22.5 nm³ membrane (70% POPC, 30% cholesterol), solvated with TIP3P water and 150 mM NaCl (with an extra four Na+ ions for neutralisation). Three independent replicates of wildtype IL-6:IL-6Rα:gp130 and IL-6:IL-6Rα:gp130$_{P498L}$ were simulated using Gromacs 2021.3[66] and the CHARMM36m forcefield[67] (detailed simulation setups in Supplementary Table 2). In each case, the system was first minimised without constraints for 500 steps with the steepest-descent integrator, before a second minimisation with hydrogen bonds fixed to atoms with the LINCS algorithm[68], until the system reached a maximum force of 1,000 kJ/mol or convergence. The resulting energy-minimised configuration was then equilibrated following the CHARMM-GUI procedure: six short runs with increasing time steps and decreasing force constants for position restraints. The first two runs were done in an NVT ensemble with a v-rescale thermostat maintaining the temperature at 310 K (τt = 1.0 ps) for 125 ps (1 fs time step). The next four runs were performed in an NPT ensemble with a semi-isotropic Berendsen barostat maintaining a pressure of 1 bar (τp = 5.0 ps) for 125 ps (1 fs time step), 250 ps (2 fs time step), 250 ps (2 fs time step) and 250 ps (2 fs time step), respectively. Finally, a production run was performed for 1 μs (2 fs time step) using the md integrator, a v-rescale thermostat (temperature of 310 K, τt = 1.0 ps) and a semi-isotropic Parinello-Rahman barostat (pressure of 1 bar, τp = 5.0 ps). The Verlet scheme was used to impose a cut-off of 1.2 nm for both van der Waals and Coulomb. The LINCS algorithm was used to constrain bonds to hydrogen atoms.

Pairwise distances between the two transmembrane helices of gp130 were measured over the course of the trajectories for residues E617, L630 and N642 in the wildtype and P498L simulations using the gmx distance command of Gromacs 2021.3. Histograms were plotted with seaborn https://joss.theoj.org/papers/10.21105/joss.03021 and Matplotlib v3.3.4 https://ieeexplore.ieee.org/document/4160265 in Python v3. The simulations reached equilibrium quickly (Supplementary Fig. 8) and, to avoid bias towards larger distances achieved in the later part of the runs, the entire production runs were used for analysis.

### Reporting summary

Further information on research design is available in the Nature Portfolio Reporting Summary linked to this article.

## Data availability

Data supporting the findings of this manuscript are available from the corresponding authors upon reasonable request. A reporting summary for this article is available as a Supplementary Information file. Source data are provided with this paper. Raw data generated in this study underlying Supplementary Fig. 6a, c and d are included as a source data file. CryoEM maps generated in this study have been deposited in the Electron Microscopy Data Bank under the accession

codes EMD-18743 (IL-11:IL-11Rα:gp130$_{P496L}$); EMD-18741 (IL-6:IL-6Rα:gp130$_{P496L}$); and EMD-18742 (IL-6:IL-6Rα:gp130). The structural models generated in this study have been deposited in the Protein Data bank under the accession codes 8QY6 (IL-11:IL-11Rα:gp130$_{P496L}$); 8QY4 (IL-6:IL-6Rα:gp130$_{P496L}$); and 8QY5 (IL-6:IL-6Rα:gp130). Structural models used to initiate model building were accessed from the Protein Data Bank under the accession code 1P9M and from the AlphaFold protein structure database entries Q00560; A8K3F7; and Q14626. Structural model used to generate Fig. 2b and Fig. 2d were accessed from the Protein Data Bank under accession code: 6O4O. Structural models used to generate Supplementary Fig. 1 were accessed from the Protein Data Bank under accession codes: 7U7N; 8D6A; 8D74; and 8D7R and from the AlphaFold protein structure database entries P40189; Q6UWB1; Q99650; P13725; P42702; P26992; and Q16619. Molecular dynamics simulation data files produced in this study are uploaded to Zenodo database https://doi.org/10.5281/zenodo.10210284. Source data are provided with this paper.

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

## Acknowledgements

We thank members of the Moraga and Bubeck laboratories for helpful advice and discussion. This project made use of time on HPC granted via the UK High-End Computing Consortium for Biomolecular Simulation, HECBioSim (http://hecbiosim.ac.uk), supported by EPSRC (grant no. EP/X035603/1). T.B.V. is funded by BBSRC Doctoral Training Program grant (BB/M011178/1). I.M. is funded by the Wellcome-Trust-202323/Z/16/Z and ERC-206-STG grant. We thank S. Islam for computational support and P. Simpson for EM support. Initial screening of samples was carried out at Imperial College London Centre for Structural Biology, cryoEM data used to calculate the final reconstruction was collected at Diamond Light Source. We thank Diamond for access and support of the Cryo-EM facilities at the UK national electron bio-imaging centre (eBIC), proposal BI25127, funded by the Wellcome Trust, Medical Research Council and BBSRC. BB/M011178/1, T.B.V; Wellcome-Trust-202323/Z/16/Z, I.M.; ERC-206-STG, I.M.

## Author contributions

Conceptualization: D.B. and I.M. Methodology: S.G., Y.J., P.K.F., T.B.V., J.S.B., E.P. Investigation: S.G., Y.J., P.K.F., T.B.V., J.S.B., E.P. Funding acquisition: D.B. and I.M. Project administration: D.B., I.M. and J.P. Supervision: D.B., I.M. and J.P. Writing-original draft: D.B. and I.M. Writing review and editing: S.G., J.P., I.M. and D.B.

## Competing interests

The authors declare no competing interests.

## Additional information

**Supplementary information** The online version contains Supplementary Material available at https://doi.org/10.1038/s41467-024-46235-6.

