## [Peer Review File · Nature Communications]

Structural insights into IL-11-mediated signalling and human IL6ST variant-associated immunodeficiencyReviewer 1: [see attached]

Reviewer Feedback for manuscript NCOMMS-23-53243

The manuscript entitled “Structural insights into IL-11-mediated signalling and human IL6ST variant-associated 3 immunodeficiency” by Gardner et al is an important study to understand the nomenclature and effect of gp130 in interleukin signaling. The authors have solved the structure of the signaling complex of IL-11R α :IL11:gp130. It is not clear why a mix of human and murine proteins are used or the sequence homology between the proteins. The authors also investigated the complex of disease mutations leading to minor changes of the complex with abolished signaling.

This is an important study conducted with background knowledge combining *in silico*, *in vitro* and *in vivo* experiments. It is a very important study to understand interleukin 11 (IL11) but also to understand chemokine signaling. I have no major concerns and believe that this paper is suitable for publication in Nature Communications after revision. I have stated my minor concerns below.

Minor edits.

1. The signalling receptor gp130 represents a paradigm for differential signal activation and functional diversity. It is required for interleukin (IL)-6, IL-11, IL-27, oncostatin-M, leukaemia inhibitory factor, ciliary neurotrophic factor, cardiotrophin-1, and cardiotrophin-like cytokine factor-1 signalling^{25, 26, 27, 28, 29}. These cytokines engage specific ligand-binding subunits (often non-signalling), but all require gp130 to control signal transduction.

– This section is hard to understand. Gp130 is the assessor receptor that plays a role in interleukin signaling and can bind to more than one interleukin ligand:receptor pair. The role of the gp130 is the assessor receptor remains elusive. Is that what this section is talking about?

2. Recent studies have reported new homozygous mutations in the IL6ST gene (encoding for gp130) that are associated with loss of function for some but not all gp130 cytokines and result in severe immune deficiencies and bone defects in patients.

- Please elaborate: What is loss of function for some but not all gp130 cytokines? Where are the mutations? Does it affect expression levels of gp130?

3. The next sentence: These gp130 mutants dramatically disrupt signalling when engaged by cytokines that homodimerize gp130, such as IL-6 and IL-11, but still retain some degree of signalling when bound by cytokines that bind gp130 in heterodimeric complexes, e.g. LIF and IL-27.

- Please explained the nomenclature of the signaling complexes discussed, i.e., why is IL6 and IIII homodimers? I(s the homodimer formed with 2 ligands or 2 receptors or 2 of each? What is the cognate pair of the heterodimer of LIF and IL-27? More information is needed to understand this introduction. A figure would be very helpful to understand the background of what is known. The sentence after this speaks about the separate ligand

binding domains. Please add a figure to explain and this section. The cognate receptor IL-11R α is not mentioned until row 82. Its stated on row 118 that figure 1a and b is showing the nomenclature of the signaling complex. Introduce Fig. 1 in the introduction for clarity for non-experts.

4. On row 85 the authors talk about “disease affected gp130 variant P498L (murine equivalent gp130P496L), and on row 97-98 the authors talk about domains D1-D6 of murine wildtype gp130 or gp130P496L (human equivalent P498L).

-Which one is it? Are you studying the human or the murine protein? Pick one nomenclature throughout the entire paper. Why is murine protein used? What is the sequence homology between human and murine proteins? What is the sequence homology in the “hotspots” found to be essential for ligand:receptor binding and/or disease mutations?

5. Row 24, 81, and 104 explains the abbreviation cryo-EM.

-Please type it out at row 24 and use the abbreviation throughout the paper.

6. *Why is a mix of human and murine proteins used?*
7. *Row 157 states beta strand while row 163 states β -strand. Please be consistent.*
8. Thus, we set to investigate whether gp130P498L mutation affects IL-6 and IL-11 binding or receptor assembly, which could account for its poor signalling. We defined the binding affinities of IL-6 and IL-11 for murine wildtype gp130 and gp130P496L via surface plasmon resonance (SPR).

-SPR is a great analysis methodology to answer these questions in comparison with in cell activity assays monitoring the phosphorylation of STAT3 versus pSTAT3. However, an SPR experiment will not provide information on the nomenclature of the complex. For example, is the on/off rate measuring binding between the 1:1 complex or the active hexameric complex? Please elaborate on the SPR results as an effect of the size of the complex measured compared to the cell assays.

Reviewer #2 (Remarks to the Author):

In their study Structural insights into IL-11-mediated signalling and human IL6ST variant associated immunodeficiency Gardner, Jin, Fyfe et al report the cryo-EM structure of the IL-11/IL-6 receptor complex. Furthermore, the authors focus on the mouse ortholog of a human gp130 mutant (P498L) that causes immunodeficiency in humans.

This study extends our insight into cytokine:receptor complexes for IL-11 and gp130-centered receptor complexes. It furthermore provides interesting insights into how a single point mutant in gp130 may compromise signaling. This being said, the conceptual advances of the study are somewhat limited in the light of recent IL-11:receptor complexes (Ref. 42 of the submitted paper) as well as gp130-centered complexes in general (Ref. 46 of the submitted paper). Furthermore, in several places the authors fall short of providing the insights they claim to provide when going beyond structural insights. To strengthen their conclusions, the authors may want to address the following points:

Major points:

1. It is true, that we have good understanding of cytokine:receptor binding on structural and biophysical terms by now, and that we are lacking the transmission step through the membrane; however, also this paper does not significantly extend our understanding of the latter in comparison to the state-of-the art.
2. The authors present data that the gp130 P->L mutants do not affect binding of hyper IL-6 / hyper IL-11; this does not rule out, however, that P->L mutants may affect binding of IL6R/IL11R to cytokine:gp130 complexes or of gp130 to IL6R/IL11R:cytokine complexes when not fixed in a single chain (and dynamics of the complexes, if altered, may impact on signaling)
3. The authors fall short of their claims in several places, e.g. in the abstract it says "By comparing our structure with that of the related IL-6 complex we discover subtle differences in binding interfaces that drive gp130 signalling outcomes." No biochemical/biological data are presented that make use of the structural insights to substantiate this statement. Analogously, the authors state "In addition, we explore how mutations in the IL6ST gene encoding for gp130, which cause severe immune deficiencies in humans, impair gp130 signalling without blocking cytokine binding." However, the presented MD data (also with no further biochemical/biological data) do not solve this question.
4. The jumping between IL-11 and IL-6 remains a bit arbitrary and makes the conceptual focus of the paper less clear.
5. In Figure 4, MD data are presented. These were done on "trimmed" AlphaFold2 models, limiting their impact, i.e.: IL-6 TMDs were removed, as they are not involved in signaling, but they may very well have an impact on the relative orientations of the gp130 TMDs. It remains unclear, how the authors interpret their data: The differences in the distance histograms for different positions within the TMDs argues for non-parallel arrangements in the membrane, as opposed to the schematics. Also, no experiments were performed based on the "naked" MD data, which could have mitigated some of the concerns.

Minor points:

1. The intro seems in places somewhat superficial, brushes over recent important findings in the field (e.g. Ref 46)
2. The P->L mutant should be indicated in Figure 4 a
3. In SI Figure 4a, errors should be given and SPR traces be shown
4. Do the MD simulations show the same differences in the ectodomain structures/dynamics like the neural network analyses?

Structural insights into IL-11-mediated signalling and human *IL6ST* variant-associated immunodeficiency

Gardner et al.

Response to reviewers

We are writing in response to your e-mail message with peer-review comments on our manuscript “Structural insights into IL-11-mediated signalling and human *IL6ST* variant-associated immunodeficiency” (NCOMMS-23-53243). We were pleased and encouraged by the reviewers’ recognition of the interest and significance of our results. We thank the reviewers for their suggestions on how we could strengthen and improve the manuscript. We have now revised our manuscript to address the reviewers’ concerns, as detailed below.

Reviewer #1 comments:

*The manuscript entitled “Structural insights into IL-11-mediated signalling and human *IL6ST* variant-associated 3 immunodeficiency” by Gardner et al is an important study to understand the nomenclature and effect of gp130 in interleukin signaling. The authors have solved the structure of the signaling complex of IL-11Ra:IL11:gp130. It is not clear why a mix of human and murine proteins are used or the sequence homology between the proteins. The authors also investigated the complex of disease mutations leading to minor changes of the complex with abolished signaling. This is an important study conducted with background knowledge combining in silico, in vitro, and in vivo experiments. It is a very important study to understand interleukin 11 (IL11) but also to understand chemokine signaling. I have no major concerns and believe that this paper is suitable for publication in Nature Communications after revision. I have stated my minor concerns below.*

We thank the reviewer for their recognition of our work.

Minor edits.

1. The signalling receptor gp130 represents a paradigm for differential signal activation and functional diversity. It is required for interleukin (IL)-6, IL-11, IL-27, oncostatin-M, leukaemia inhibitory factor, ciliary neurotrophic factor, cardiotrophin-1, and cardiotrophin-like cytokine factor-1 signalling^{25, 26, 27, 28, 29}. These cytokines engage specific ligand-binding subunits (often non-signalling), but all require gp130 to control signal transduction.

– This section is hard to understand. Gp130 is the assessor receptor that plays a role in interleukin signaling and can bind to more than one interleukin ligand:receptor pair. The role of the gp130 is the assessor receptor remains elusive. Is that what this section is talking about?

Response 1: This section is intended to provide a basic introduction to gp130 signalling. For clarity we have now expanded this paragraph and included a new supplementary figure (Supplementary Fig. 1, revised numbering) to show the different binding partners of gp130 and highlight complexes that signal through homodimeric gp130 (IL-6 and IL-11 complexes) or when gp130 signals through heterodimerization with a co-receptor (IL-27, oncostatin-M, leukaemia inhibitory factor, ciliary neurotrophic factor, cardiotrophin-1, and cardiotrophin-like cytokine factor-1).

*2. Recent studies have reported new homozygous mutations in the *IL6ST* gene (encoding for gp130) that are associated with loss of function for some but not all gp130 cytokines and result in severe immune deficiencies and bone defects in patients.*

- Please elaborate: What is loss of function for some but not all gp130 cytokines? Where are the mutations? Does it affect expression levels of gp130?

Response 2: We have revised the introduction to provide more information on the gp130 mutations identified in patients with a recessive form of hyper-IgE syndrome described in the study by Chen et

al., 2021 (DOI: 10.1016/j.jaci.2021.02.044). This study reports wildtype levels of surface expression for P498L and N404Y mutants in cells. Although the A517P variant was not detected on the surface, immunoblot analysis showed that all three disease-associated variants were expressed at levels similar to wildtype. The positions of these point mutations (N404Y, P498L and A517P) are indicated in the schematic shown in the new Supplementary Fig. 1 and a brief description of their effects on IL-11, IL-6, IL-27 and LIF signalling are now included in the introduction.

3. The next sentence: These gp130 mutants dramatically disrupt signalling when engaged by cytokines that homodimerize gp130, such as IL-6 and IL-11, but still retain some degree of signalling when bound by cytokines that bind gp130 in heterodimeric complexes, e.g. LIF and IL-27.

- Please explained the nomenclature of the signaling complexes discussed, i.e., why is IL6 and IL11 homodimers? Is the homodimer formed with 2 ligands or 2 receptors or 2 of each? What is the cognate pair of the heterodimer of LIF and IL-27? More information is needed to understand this introduction. A figure would be very helpful to understand the background of what is known. The sentence after this speaks about the separate ligand binding domains. Please add a figure to explain and this section. The cognate receptor IL-11R α is not mentioned until row 82. Its stated on row 118 that figure 1a and b is showing the nomenclature of the signaling complex. Introduce Fig. 1 in the introduction for clarity for non-experts.

Response 3: A cartoon diagram showing gp130 binding complexes is now included in Supplementary Fig. 1. We have clarified the introduction to explain what is meant by homodimeric and heterodimeric gp130 signalling complexes and reinforce this message in the cartoon diagram. The diagram includes details for the cognate pair in LIF and IL-27 signaling complexes. In doing, so we now mention IL-11R α much earlier in the introduction.

4. On row 85 the authors talk about “disease affected gp130 variant P498L (murine equivalent gp130P496L), and on row 97-98 the authors talk about domains D1-D6 of murine wildtype gp130 or gp130P496L (human equivalent P498L).

-Which one is it? Are you studying the human or the murine protein? Pick one nomenclature throughout the entire paper. Why is murine protein used? What is the sequence homology between human and murine proteins? What is the sequence homology in the “hotspots” found to be essential for ligand:receptor binding and/or disease mutations?

Response 4: Initially we tried to purify IL-6 and IL-11 complexes with human gp130; however, our yields were too low especially for the disease associated variant. Given the high sequence conservation between murine and human gp130, we used murine gp130 as a proxy for the human protein in cryoEM and SPR experiments that required purified recombinant proteins. We have now included a sequence alignment in the Supplementary material (Supplementary Fig. 2, revised numbering) showing the conservation between *murine* and *human* gp130 and highlighting interface residues in our cryoEM structure.

We used human gp130 for our MD simulations and cellular assays that required interactions with downstream signalling proteins. Our signalling and single molecule imaging used a HeLa cell line which lacked gp130 (Martinez-Fabregas et al., 2019; doi: 10.7554/eLife.49314) to ensure that results generated were not influenced by background levels of endogenous protein.

We have modified the text to clarify the rationale for when murine or human gp130 was used for each type of experiment and included a new panel in Supplementary Fig. 2, revised numbering) to show the sequence homology between the proteins.

5. Row 24, 81, and 104 explains the abbreviation cryo-EM.
-Please type it out at row 24 and use the abbreviation throughout the paper.

Response 5: This is now done.

6. Why is a mix of human and murine proteins used?

Response 6: Due to issues of yield when trying to purify the human gp130 variant, murine gp130 was used in experiments that required purification of recombinant proteins including cryoEM and SPR experiments. Human gp130 was used for cellular-based experiments and molecular dynamics simulations. See Response 4 for more details.

7. Row 157 states beta strand while row 163 states b-strand. Please be consistent.

Response 7: This is now done.

8. Thus, we set to investigate whether gp130P498L mutation affects IL-6 and IL-11 binding or receptor assembly, which could account for its poor signalling. We defined the binding affinities of IL-6 and IL-11 for murine wildtype gp130 and gp130P496L via surface plasmon resonance (SPR).
-SPR is a great analysis methodology to answer these questions in comparison within cell activity assays monitoring the phosphorylation of STAT3 versus pSTAT3. However, an SPR experiment will not provide information on the nomenclature of the complex. For example, is the on/off rate measuring binding between the 1:1 complex or the active hexameric complex? Please elaborate on the SPR results as an effect of the size of the complex measured compared to the cell assays.

Response 8: Interactions between monovalent surface-bound ligands and multivalent analytes such as antibodies generate avidity effects that are difficult to analyse. Thus, in this context it is advised to immobilize the multivalent molecule on the surface of the SPR chip and inject its binding partner as the analyte. We and others have followed this principle when analysing IL-6 and IL-11 binding affinities for gp130 and obtained comparable K_D measurements (doi.org/10.7554/eLife.68843; doi/10.4049/jimmunol.1402908; doi: 10.1038/srep37716; doi: 10.7554/eLife.49314). Gp130 was immobilized on the chip surface and IL-6 and IL-11 were used as analytes. In this context, we are not making claim about the stoichiometry of the IL-6 and IL-11 complex, but measuring 1:1 interaction between gp130 and IL-6 and IL-11 as denoted by the 1:1 kinetic fitting model that we used to obtain the binding constants. We are now also presenting the equilibrium fitting binding constants obtained from our SPR studies as additional evidence that gp130_{P496L} mutant binds with comparable affinity to IL-6 and IL-11 as gp130_{WT}.

Reviewer #2 (Remarks to the Author):

In their study Structural insights into IL-11-mediated signalling and human IL6ST variant associated immunodeficiency Gardner, Jin, Fyfe et al report the cryo-EM structure of the IL-11/IL-6 receptor complex. Furthermore, the authors focus on the mouse ortholog of a human gp130 mutant (P498L) that causes immunodeficiency in humans.

This study extends our insight into cytokine:receptor complexes for IL-11 and gp130-centered receptor complexes. It furthermore provides interesting insights into how a single point mutant in gp130 may compromise signaling. This being said, the conceptual advances of the study are somewhat limited in the light of recent IL-11:receptor complexes (Ref. 42 of the submitted paper) as well as gp130-centered complexes in general (Ref. 46 of the submitted paper). Furthermore, in

several places the authors fall short of providing the insights they claim to provide when going beyond structural insights. To strengthen their conclusions, the authors may want to address the following points:

Major points:

1. It is true, that we have good understanding of cytokine:receptor binding on structural and biophysical terms by now, and that we are lacking the transmission step through the membrane; however, also this paper does not significantly extend our understanding of the latter in comparison to the state-of-the art.

Response 1: Reference 46 of our submitted manuscript, citing the gp130-centered structures, notably lacked the complex with IL-11 and its co-receptor. Our study provides the structural basis for IL-11 recognition by gp130 and IL-11R α and is a key piece of the puzzle in understanding selectivity in gp130-mediated signalling. Reference 42 refers to a non-peer reviewed pre-print of the IL-11:receptor complex, which has been published in Nature Communications after the submission of our manuscript. It is the editorial policy of Nature Communications that any similar papers published independently after submission will not compromise the novelty of our study. We have updated the reference to now cite the published work. Finally, our data provide the first experimentally determined structures of complexes with gp130 disease-related variants.

2. The authors present data that the gp130 P->L mutants do not affect binding of hyper IL-6 / hyper IL-11; this does not rule out, however, that P->L mutants may affect binding of IL6R/IL11R to cytokine:gp130 complexes or of gp130 to IL6R/IL11R:cytokine complexes when not fixed in a single chain (and dynamics of the complexes, if altered, may impact on signaling).

Response 2: There is a precedent in the literature for using cytokine fusion constructs to characterise signalling defects of the hyper-IgE syndrome associated gp130 variants, including the P->L mutant which we focus on here (DOI:<https://doi.org/10.1016/j.jaci.2021.02.044>). Our phospho-flow analysis is consistent with previously published results showing impaired STAT3 signalling of gp130_{P498L} when stimulated by either IL6 or IL11 fusion constructs. Throughout the rest of the manuscript, we investigate the underlying cause of this observed reduction in signalling using SPR, single-molecule imaging and cryoEM analyses. We acknowledge that there may be effects on dynamics or binding kinetics that could additionally impact signalling when the cytokine is not fixed in a single chain with its co-receptor. We have added a statement in the text to reflect this.

3. The authors fall short of their claims in several places, e.g. in the abstract it says “By comparing our structure with that of the related IL-6 complex we discover subtle differences in binding interfaces that drive gp130 signalling outcomes.” No biochemical/biological data are presented that make use of the structural insights to substantiate this statement.

Response 3a: Mutation in *human* gp130 R281Q selectively impairs signalling by IL-11 but not by IL-6 (<https://doi.org/10.1038/s41413-020-0098-z>). Our cryoEM data shows how the local environment of the equivalent *murine* residue differs between the IL-11 and IL-6 structures, providing a structural basis for the signalling selectivity previously reported. Although this residue is conserved between *mouse* and *human*, we have softened the language in our abstract and provided an additional supplemental figure clearly showing the conservation of interface residues between the two species.

Analogously, the authors state “In addition, we explore how mutations in the IL6ST gene encoding for gp130, which cause severe immune deficiencies in humans, impair gp130 signalling without blocking cytokine binding.” However, the presented MD data (also with no further biochemical/biological data) do not solve this question.

Response 3b: In the abstract quote above, we only state that “we explore how” the P498L mutation impairs signalling without blocking cytokine binding. Our structural data show that complexes formed from the mutant gp130 are more flexible which could provide a possible explanation for the reduced signalling. We have now softened the language the second to last paragraph of the discussion along these lines.

4. The jumping between IL-11 and IL-6 remains a bit arbitrary and makes the conceptual focus of the paper less clear.

Response4: IL-11 and IL-6 are the only two members of the gp130 cytokine family that require dimerization of gp130 for signalling. By including data for both complexes we provide a more complete understanding of this family of cytokines. In the revised manuscript we include a supplementary figure in the introduction (Supplementary Fig. 1, revised numbering) to make it more clear for the non-expert how IL-6 and IL-11 complexes compare to other gp130-binding cytokines.

5. In Figure 4, MD data are presented. These were done on “trimmed” AlphaFold2 models, limiting their impact, i.e.: IL-6 TMDs were removed, as they are not involved in signaling, but they may very well have an impact on the relative orientations of the gp130 TMDs.

Response 5a: Only domains D2 and D3 of IL-6R α are required for signalling when fused in a single chain with IL-6. In addition, these were the domain boundaries for IL-6R α in our cryoEM structures. To directly assess the effect of the lipid bilayer on the orientation of the gp130 transmembrane helices, we only included this one extra component. We agree with the reviewer that the full-length co-receptor could additionally influence the orientation of gp130 and have included a statement in the text to reflect this.

It remains unclear, how the authors interpret their data: The differences in the distance histograms for different positions within the TMDs argues for non-parallel arrangements in the membrane, as opposed to the schematics.

Response 5b: We agree that the schematic representation of the helices maybe misleading. To provide clarity, we now show the location of those residues in the context of the full membrane environment used in the simulation. E617, L630 and N642 are now labelled in Figure 4a.

Also, no experiments were performed based on the “naked” MD data, which could have mitigated some of the concerns.

Response 5c: Our *in silico* analysis informs new testable hypotheses that will frame future biochemical and biophysical investigations of gp130 function. While we agree with the reviewer that follow-up experimental analysis will further advance the field, that is beyond the scope of this study.

Minor points:

1. The intro seems in places somewhat superficial, brushes over recent important findings in the field (e.g. Ref 46)

Response 6: We have updated the introduction to provide more depth and background on gp130 complexes and the effects of disease-related variants. See responses 1-3 for Reviewer 1 for details.

2. The P->L mutant should be indicated in Figure 4 a

Response 7: This is now done.

3. In SI Figure 4a, errors should be given and SPR traces be shown

Response 8: This is now done. The errors are now given and SPR traces are now shown in Supplementary Fig. 5 (revised numbering). The legend is also updated to reflect this.

4. Do the MD simulations show the same differences in the ectodomain structures/dynamics like the neural network analyses?

Response 8: We performed flexible fitting of the IL-6 wildtype and mutant structures into the maps from the 3D flex analysis. Because of the low resolution of the final domains, we used distance restraints in Isolde on the gp130 domains. We then measured the mean distances between C α carbons of V588 as a proxy for domain 6. We compared the mean distances and standard deviations between the pairs of C α carbons. We performed a similar analysis for the equivalent position in the MD simulations which included transmembrane helices of gp130 dimers. While the distance pairs were similar in both types of analyses for the wild type, they differed for the mutant pairs. Therefore, we conclude that we cannot directly compare the ectodomain structures/dynamics because of the impact of the membrane on the system. We have removed a statement in the results section along these lines.

Reviewer #1 (Remarks to the Author):

The updated manuscript entitled "Structural insights into IL-11-mediated signalling and human IL6ST variant-associated immunodeficiency" has significantly improved and clarified the major concerns addressed by the reviewers.

Reviewer #2 (Remarks to the Author):

In their revision, the authors have addressed most of my concerns, although without providing any further data that would have substantiated their claims. This remains a weak point of the paper, that some findings remain un-substantiated by further experiments (e.g. for the MD findings). This being said, with the adjustments of the text, the paper should now be publishable.

Reviewer #3 (Remarks to the Author):

Gardner et al. determined the structures of IL-11 and IL-6 complexes with the gp130 mutant that causes human disease using cryo-EM. Using a motion-based deep neural network, the authors suggest that increased flexibility of the ectodomain caused by the mutation (i.e. gp130 P496L) alters the distance between transmembrane helices near the extracellular face of the membrane, which affects the receptor downstream signaling. This is an important study which explores how a single mutation alter chemokine signaling without directly affecting the ligand binding. Before considering publication, it is necessary to address the following points:

1. There is a discrepancy between Supplementary Table 1 and the PDB report. Please clarify these inconsistencies.
2. Figure 1 demonstrates the structure of the IL-11/IL-11Ra complex with gp130 mutant (P496L), not wild-type gp130. This information should be clearly indicated in Figure and Figure legend. Further, the residue (L496 in gp130) should be shown in the main figure or supplementary figure if the authors can see it in their model. As the model of IL-11/IL-11Ra/wild-type gp130 complex is available, it would be beneficial to provide a structural comparison between the wild-type and mutant, especially for the residue Leu496 (Pro496 in mutant).
3. The domain information (e.g., D1-D6) should be noted in the overall structure as well as zoom in on the specific sites of the IL-11/IL-11Ra/gp130 mutant complex.
4. Line 133: "Upon complex formation, a loop within IL-11 (residues T77-L90) at Site 1 becomes restructured to form a short helix that nestles in the groove created by the hinge of IL-11Ra (Fig. 2).": With the current Fig. 2, it is difficult to follow the description. The figure and text should be revised.
5. Line 145: "thus we conclude that any potential differences due to the chimera were negligible.": This chimera indicates that IL-11 (or IL-6) has been fused with its co-receptor through a short linker? Despite agreeing with the author's argument, the lack of a difference in IL-6 complex with a similar method does not imply that IL-11/IL-11Ra/gp130 complex will not be altered by fusion. It is necessary to revise this conclusion. The manuscript should further describe the structural observations of the linker region. For example, was the short linker determined in the structure? Does it interact with any region of IL-11Ra or gp130? Was it disordered due to its flexibility?
6. Line 154: "By contrast, gp130:E273 points towards a tryptophan (W214) in the IL-6 receptor recognition complex (Fig. 3b)." In Figure 3b for IL-6 Site 2b, gp130:E273 points toward IL-6Ra:W233. Please clarify this.

Structural insights into IL-11-mediated signalling and human *IL6ST* variant-associated immunodeficiency

Gardner et al.

Response to reviewers

We are writing in response to your e-mail message with peer-review comments on our manuscript “Structural insights into IL-11-mediated signalling and human *IL6ST* variant-associated immunodeficiency” (NCOMMS-23-53243A). We are pleased that the concerns of Reviewers 1 and 2 have been satisfied. We have now revised our manuscript to address the technical clarifications requested by Reviewer 3.

REVIEWERS' COMMENTS

Reviewer #1 (Remarks to the Author):

*The updated manuscript entitled “Structural insights into IL-11-mediated signalling and human *IL6ST* variant-associated immunodeficiency” has significantly improved and clarified the major concerns addressed by the reviewers.*

Reviewer #2 (Remarks to the Author):

In their revision, the authors have addressed most of my concerns, although without providing any further data that would have substantiated their claims. This remains a weak point of the paper, that some findings remain un-substantiated by further experiments (e.g. for the MD findings). This being said, with the adjustments of the text, the paper should now be publishable.

Reviewer #3 (Remarks to the Author):

Gardner et al. determined the structures of IL-11 and IL-6 complexes with the gp130 mutant that causes human disease using cryo-EM. Using a motion-based deep neural network, the authors suggest that increased flexibility of the ectodomain caused by the mutation (i.e. gp130 P496L) alters the distance between transmembrane helices near the extracellular face of the membrane, which affects the receptor downstream signaling. This is an important study which explores how a single mutation alter chemokine signaling without directly affecting the ligand binding. Before considering publication, it is necessary to address the following points:

1. There is a discrepancy between Supplementary Table 1 and the PDB report. Please clarify these inconsistencies.

We thank the reviewer for their careful reading of our manuscript. Indeed, there was a mismatch in the statistics for the IL-11 model stemming from entering the values from the wrong phenix validation job. These were minor and have now been amended in a revised Supplementary Table 1. There are a few other minor discrepancies between our table and the PDB report which may be due to differences in software used. For transparency, we state the software version numbers we used to generate our table in the Reporting Summary.

2. Figure 1 demonstrates the structure of the IL-11/IL-11Ra complex with gp130 mutant (P496L), not wild-type gp130. This information should be clearly indicated in Figure and Figure legend. Further, the residue (L496 in gp130) should be shown in the main figure or

supplementary figure if the authors can see it in their model. As the model of IL-11/IL-11Ra/wild-type gp130 complex is available, it would be beneficial to provide a structural comparison between the wild-type and mutant, especially for the residue Leu496 (Pro496 in mutant).

We have now modified the image (Figs 1 and 2) and legend make it clear that the gp130 used in this structure was the P496L variant.

Density for this residue is not strong in our high-resolution sharpened map; therefore, we only indicate the C α carbon position of this residue in the context of the D1-D6 gp130_{P496L} model (Figure 1b). The model from the published wildtype structure is limited to domains 1-3 of gp130 and does not include residue 496.

3. The domain information (e.g., D1-D6) should be noted in the overall structure as well as zoom in on the specific sites of the IL-11/IL-11Ra/gp130 mutant complex.

We now include labels for domains D1-D6 on the full model shown in Figure 1b. As the site positions are already indicated on the full model, we feel that it would overcrowd the zoomed in panels which focus on sidechain interactions.

4. Line 133: "Upon complex formation, a loop within IL-11 (residues T77-L90) at Site 1 becomes restructured to form a short helix that nestles in the groove created by the hinge of IL-11R α (Fig. 2).": With the current Fig. 2, it is difficult to follow the description. The figure and text should be revised.

We have now clarified this in the text and legend.

5. Line 145: "thus we conclude that any potential differences due to the chimera were negligible.": This chimera indicates that IL-11 (or IL-6) has been fused with its co-receptor through a short linker? Despite agreeing with the author's argument, the lack of a difference in IL-6 complex with a similar method does not imply that IL-11/IL-11Ra/gp130 complex will not be altered by fusion. It is necessary to revise this conclusion. The manuscript should further describe the structural observations of the linker region. For example, was the short linker determined in the structure? Does it interact with any region of IL-11Ra or gp130? Was it disordered due to its flexibility?

The word chimera refers to a complex formed from a mixture of murine gp130 and human cytokine-fusions. A cytokine-fusion refers to the single polypeptide chain in which IL-6 and IL-11 cytokines are fused by a short linker to the second and third domains of their co-receptors, IL-6R α and IL-11R α respectively (Supplementary Fig. 2a). This linker is highly flexible and is disordered in our structures. To rule out that any differences we might observe in the IL-11 complex were due to using a human/mouse chimera complex, we solved the human/mouse chimeric IL-6:IL-6R α : murine gp130 complex and compared it to the published IL-6:IL-6R α :gp130 complex comprised of all human proteins. This is now made clearer in the text.

6. Line 154: "By contrast, gp130:E273 points towards a tryptophan (W214) in the IL-6 receptor recognition complex (Fig. 3b)." In Figure 3b for IL-6 Site 2b, gp130:E273 points toward IL-6Ra:W233. Please clarify this.

The correct residue is W233. This is now changed in the text.